# HLR-SQL: Human-Like Reasoning for Text-to-SQL

Timo Eckmann
timo.eckmann@tu-darmstadt.de
Technical University of Darmstadt
Darmstadt, Germany

Matthias Urban
matthias.urban@tu-darmstadt.de
Technical University of Darmstadt
Darmstadt, Germany

Jan-Micha Bodensohn
jan-micha.bodensohn@tu-darmstadt.de
DFKI & Technical University of Darmstadt
Darmstadt, Germany

Carsten Binnig
carsten.binnig@tu-darmstadt.de
Technical University of Darmstadt & DFKI
Darmstadt, Germany

## Abstract

Recent LLM-based approaches have achieved impressive results on Text-to-SQL benchmarks such as Spider and Bird. However, a key limitation of these benchmarks is that their queries do not reflect the complexity typically seen in real-world enterprise scenarios. In this paper, we introduce HLR-SQL, a new approach designed to handle such complex enterprise SQL queries. Unlike existing methods, HLR-SQL imitates Human-Like Reasoning with LLMs by incrementally composing queries through a sequence of intermediate steps, gradually building up to the full query. We evaluate HLR-SQL on a newly constructed benchmark, Spider-HJ, which systematically increases query complexity by splitting tables in the original Spider dataset to raise the average join count needed by queries. Our experiments show that state-of-the-art models experience up to a 70% drop in execution accuracy on Spider-HJ, while HLR-SQL achieves a 9.51% improvement over the best existing approaches on the Spider leaderboard.

**ACM Reference Format:**
Timo Eckmann, Matthias Urban, Jan-Micha Bodensohn, and Carsten Binnig. 2025. HLR-SQL: Human-Like Reasoning for Text-to-SQL. In *Novel Optimizations for Visionary AI Systems (NOVAS '25), June 22–27, 2025, Berlin, Germany*. ACM, New York, NY, USA, 5 pages. https://doi.org/10.1145/3735079.3735323

## 1 Introduction

**LLMs dominate Text-to-SQL.** Text-to-SQL, the task of translating natural language questions into SQL statements, has recently gained traction because it enables non-expert users to query databases, substantially enlarging their user bases. Recent state-of-the-art approaches employ Large Language Models (LLMs) with prompt engineering [4, 6, 12, 13] or supervised fine-tuning [10] and achieve impressive translation accuracies on established academic benchmarks such as Spider [16] and Bird [11]. For example, DAIL-SQL [6], currently the best publicly listed approach on Spider, can correctly translate 86.6% of queries in the Spider test set.

**Real-world queries are more complex.** Existing benchmarks, however, do not fully reflect the complexity of real-world settings. For instance, as reported in [1], real-world databases are often highly normalized, causing even simple natural language questions to require 4.01 joins per query on average. In contrast, queries in the Spider benchmark [16] require only 0.5 joins on average, and those in Bird [11] only around 1 join (i.e., 2 tables). More importantly, only 60 out of 7000 queries in Spider and 74 in Bird (approximately 0.85% and 0.78% of the respective training sets) require four or more joins. This low maximum join count demonstrates that existing benchmarks do not fully capture the real world with SQL queries that involve many join operations.

**Text-to-SQL for complex queries.** In this paper, we thus set out the goal of studying Text-to-SQL in scenarios that require a high number of join clauses. As we show in our evaluation, existing approaches leading classical benchmarks fail on such complex queries. To solve them, we argue that Text-to-SQL systems must approach the task of SQL generation more like humans do. When a human is tasked with formulating a complex SQL statement, they typically do not write the whole statement in one go. Instead, they start with a simpler statement like *SELECT \* FROM <tablename>*, which they might even execute. Based on this initial statement, we then iteratively make it more complex or combine multiple simple SQL statements using joins or nested queries.

**Human-Like Reasoning for Text-to-SQL.** Based on this idea, we propose a novel approach to emulate human reasoning when solving the Text-to-SQL task through iterative composition, which involves actions such as note-taking, query refinement, database exploration, and error correction. In this paper, we present a first prototype called HLR-SQL, which implements this idea on top of an LLM that uses an external memory to keep track of the artifacts of the incremental query construction. An important aspect of HLR-SQL is that rather than following a fixed procedure, it operates with complete autonomy and may adopt different actions to incrementally refine a query. As such, HLR-SQL autonomously revises and combines queries from previous steps until it determines that a satisfactory solution has been found.

**A more realistic benchmark is needed.** To better understand the challenges arising from more complex queries with higher join counts for the Text-to-SQL setting, we propose *Spider-HJ*, a novel dataset derived from Spider [16]. Unlike Spider, where queries require only 0.5 joins on average, and only 60 queries require four or more joins, *Spider-HJ* features queries with an average of 5.64

joins per query and a maximum of 20 joins. To obtain such a dataset, we keep the high-quality natural language questions of Spider and modify only the database schema in a way that requires more joins to answer the questions. Thus, any performance decrease can be directly attributed to the increased complexity of the underlying schema. In this way, *Spider-HJ* enables a precise assessment of a system's resilience to an increase in necessary join operations.

**Initial results.** Our initial evaluation shows that increased query complexity leads to a sharp decline in the execution accuracy of current state-of-the-art approaches. For example, DAIL-SQL [6], which is top-ranked on the Spider leaderboard, drops from 86.6% on the Spider test set to just 15.14% on our dataset. In contrast, HLR-SQL significantly improves over these approaches in accuracy by approximately 10%, highlighting the benefits of HLR-SQL while also making it clear that more research is needed.

## 2 Human-Like Reasoning for Text-To-SQL

In this section, we present our vision of Text-to-SQL pipelines that reason like humans do. Furthermore, we present our initial prototype HLR-SQL that mimics this Human-Like Reasoning.

### 2.1 The Need for a New Approach

Modern Text-to-SQL approaches like DIN-SQL [13], DAIL-SQL [6], and MAC-SQL [14] achieve strong performance but suffer from a fundamental limitation: they typically rely on a fixed, predefined sequence of generation steps. This restriction contrasts sharply with how humans naturally use multiple steps to solve complex tasks. In the case of Text-to-SQL, humans craft complex queries by starting with simple queries, iteratively refining them, observing feedback from the database by executing these simpler queries, and then incrementally improving them. Naturally, this process will take more iterations for complex SQL statements and thus be longer than for simple statements. Especially queries involving multiple joins and nested sub-queries over large-scale databases with many tables benefit from such an iterative, human-like approach that automatically adapts the amount of "effort" (i.e., compute and number of database interactions) to the complexity of the question at hand.

Some recent approaches, such as CHASE-SQL [12] and XiYan-SQL [7], increase the amount of effort independently of the question complexity by employing ensemble methods that generate multiple sample solutions and then choose a final query using a specially tuned selection model. MAC-SQL [14], on the other hand, tries to first estimate question complexity and then decomposes questions accordingly into multiple sub-questions. These sub-questions are then translated independently into SQL sub-queries and combined to obtain the final SQL. Unlike HLR-SQL, they do not incrementally build queries and test sub-queries by executing them. As such, wrong assumptions or mistakes early in the reasoning chain (i.e., in the formulation of early sub-queries) might lead to error propagation, resulting in final SQL statements that are vastly different from the ground truth and, thus, hard to fix.

For example, consider the query "Find the name of instructors who taught in Fall 2009 but not in Spring 2010" in a database where the instructor name is stored in an *instructors* table, and the *courses* table stores information about when they taught. A potential first sub-question could be "Which instructor taught in Fall 2009?" which

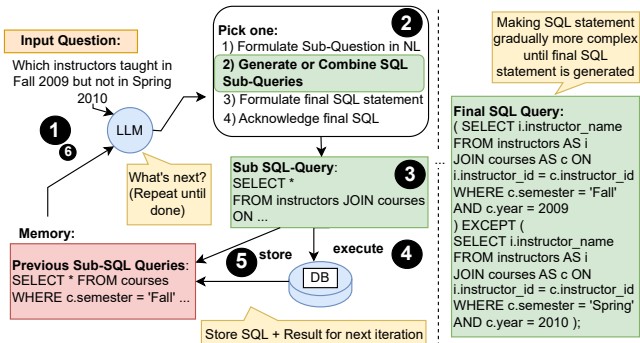

**Figure 1: HLR-SQL creates SQL statements by iteratively combining and revising SQL sub-queries. The procedure starts by putting the input question and the content of the memory (initially empty) in a prompt ①. Note that the figure shows a later iteration where a sub-SQL from a previous iteration is already in the memory. Given the prompt, the LLM decides ② to combine that existing query with a new query, which it generates ③. The generated query is run on the database ④, and the result is stored in the memory ⑤. This process of iterative refinement repeats ⑥ until the LLM acknowledges the final SQL query (right side) to be correct.**

already involves a join and can thus fail when the wrong join key is used. In this case, MAC-SQL will be oblivious to the mistake as it does not check sub-queries. In contrast, a human would first verify the list of fall instructors by executing a respective SQL query on the database and fixing it until it works, then separately identifying those in Spring, and finally combining the two results in a larger query, thereby reducing the likelihood of such error propagation.

### 2.2 Our Approach: HLR-SQL

HLR-SQL tries to mimic the human-like, iterative problem-solving approach to SQL query composition. Instead of committing to a single static pipeline, an LLM agent refines queries based on intermediate feedback, splitting a complex problem into smaller tasks and revising incorrect assumptions repeatedly along the way. We call this cycle the *draft, test, learn, and improve* process.

**Key aspects of HLR-SQL.** Most importantly, the agent in HLR-SQL is completely *autonomous* in its iterative Text-to-SQL construction. This means that it can terminate or continue depending on its assessment of the problem's complexity and the quality of its current solution. Nevertheless, to prevent endless loops, HLR-SQL can only re-iterate until a maximum of iterations is reached. Another key aspect of HLR-SQL is its *memory*, which tracks all information about previous steps and sub-results that have been obtained so far. To achieve complete autonomy, in each step, we provide the LLM with the question, the available data, and its memory from previous steps and let it decide which next steps it wants to take.

**Composition process in HLR-SQL.** Figure 1 shows the behavior of HLR-SQL using a simple example. Denoted as ①, the user submits a natural language question to the agent, which triggers the incremental query construction. In Figure 1, there is already a previous SQL sub-query present in the memory, which resulted from a previous iteration. In step ②, HLR-SQL picks one of four capabilities to refine this query: *(1) Formulate Sub-Question in NL:*

The LLM formulates a sub-question in natural language by breaking down the input question into sub-question(s) similar to chain-of-thought prompting [15]. *(2) Generate or combine SQL sub-queries:* Here, the LLM either generates completely new SQL fragments or uses the existing SQL sub-queries from memory and combines them with a new sub-query, which it generates. *(3) Formulate final SQL statement:* The LLM chooses this capability if it intends to create the final SQL query that solves the original Text-to-SQL problem it was presented with. *(4) Acknowledge final SQL:* As a last capability, HLR-SQL also provides the option to acknowledge the final SQL query. This step terminates the overall procedure.

If the agent selects capability (2) or (3), the agent will output a SQL string and execute it over the database (see ③+④ in Figure 1). A sample of the query execution result is put into the memory of the agent for further refinement. In our example in Figure 1, the LLM chooses to combine the existing SQL sub-query with a new sub-query that joins the *instructors* and *courses* tables, which is then executed on the actual database (step ④) and the result stored in the memory (step ⑤). After multiple iterations, HLR-SQL will then output and acknowledge the results of the final SQL query (see right side of Figure 1).

**Autonomy and error recovery.** A benefit of HLR-SQL is the self-reliance of the agent when exploring different parts of the problem as it sees fit. This allows it to fix assumption errors that go beyond simple SQL syntax errors. An example of this might be a synonym mismatch where the user asks for "films," but the column is labeled "movies." Our agent can test both columns and adapt based on the obtained sub-results.

## 3 Spider-HJ: A Benchmark with Many Joins

Existing benchmarks are great resources for advancing Text-to-SQL research [3, 5]. Yet, they so far have less of a focus on complex queries that require many joins (see Table 1). In this paper, we thus present a new benchmark called *Spider-HJ*, which builds on Spider [16] but contains queries with increasing SQL complexity in terms of the number of joins. We leverage the existing NL-to-SQL pairs from the Spider dataset and introduce schema modifications that necessitate additional join operations. As such, we can accurately reason that any performance differences are due to the increase in the number of required joins and the schema complexity.

**Constructing a dataset with many joins.** To increase the total number of joins required for a given question, we adopt a multi-stage process that transforms the original database schema from Spider along with the SQL queries into several variants all corresponding to the same natural language question. Each of these variants has an increased number of joins. For example, if a query selects columns $c_1, c_2$ of table $T$, we create three variants: one that splits off $c_1$ (introducing one extra join), one that splits off $c_2$ using another join key (introducing one extra join), and one that splits off both columns from the original table (introducing two extra joins).

The steps of our benchmark construction process are as follows: *(1) Parsing the original SQL query:* Each SQL query is first parsed to identify the specific columns used in its execution. This step ensures that splitting these columns necessitates joins. *(2) Partitioning and augmenting the used columns:* The columns identified are separated into new tables. For each column, a corresponding table

| Dataset | Avg. # joins | Max. # joins |
|---|---|---|
| Spider | 0.54 | 8 |
| Bird | 1.02 | 6 |
| *Spider-HJ* (ours) | **5.64** | **20** |

**Table 1: Comparison of Spider, Bird, and *Spider-HJ* regarding join counts. Our dataset contains 10 times more joins on average than Spider and also a 2.5 times increase in the maximum join counts found in the benchmark.**

is created. This process can lead to unrealistically many small tables that only contain foreign keys and the queried column. To make sure that the resulting tables have a realistic number of columns, we append a random number of columns (between 3-5) that are generated by an LLM. *(3) Random column selection and processing:* Since we only split columns that are required by the query into new tables, this makes it very obvious which tables must be joined. Therefore, we randomly select up to 5 additional columns from the original schema — even if they are not directly referenced by the query — and subject them to a similar partitioning process. These columns are also allocated to new tables and padded with three to five LLM-generated columns, thereby making it less obvious for an LLM that split tables are needed in the golden query. *(4) Generating multiple schema variants:* For each base SQL query, we systematically generate all possible combinations of column splits. This results in multiple schema variants corresponding to the same natural language question, thereby increasing the number of required join operations and introducing nuanced variations in the database schema. We repeat this process to incrementally add more joins.

**Characteristics of *Spider-HJ*.** By adopting this approach, we generate more than 20,000 questions with an average join count of 5.64 joins per question. As shown in Table 1, this is more than 10 times the average joins per question of Spider and more than 5 times in comparison to Bird. A benefit of this approach is that we can make use of the quality of the existing natural language questions of Spider, only increasing the complexity of the schema and SQL query without changing the question complexity.

## 4 Initial Experimental Evaluation

In this section, we present our initial evaluation of HLR-SQL on *Spider-HJ*. Our initial results indicate that it can better adapt to the more complex schemata than other recent Text-to-SQL approaches.

### 4.1 Experiment Setup

**Dataset.** To examine the effect of varying the number of joins, we use the previously-introduced *Spider-HJ* dataset and randomly sample 400 distinct queries for the different join counts present in *Spider-HJ*. For the join counts 16, 18, and 20, there are only 381, 301, 120 questions in Spider-HJ. Therefore, we include all of them.

**Metric.** In line with other Text-to-SQL research, we employ execution accuracy (**EX**) as our primary metric. A generated query is considered as correct if its execution result matches that of the golden SQL query, regardless of differences in query formulation.

**Models.** We use **GPT-4o-Mini-2024-07-18** because it is considerably more cost-efficient than GPT-4o while still demonstrating remarkable performance on the original Spider test set [9].

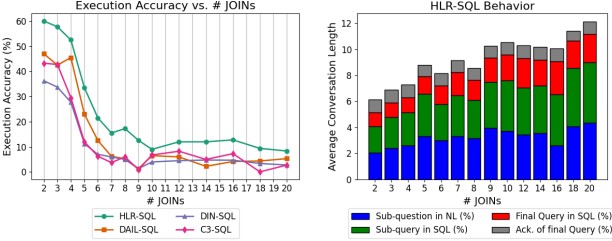

**(a) HLR-SQL vs Baselines**    **(b) HLR-SQL: # of Steps**

**Figure 2: Subplot (a) shows the accuracy on *Spider-HJ*. Subplot (b) shows the amount and type of iterative steps of HLR-SQL.**

**Baselines.** As baselines, we use the top three publicly available and listed approaches on the Spider leaderboard [16]: **DIN-SQL**[13], **DAIL-SQL**[6], and **C3-SQL**[4]. We slightly adapted all approaches to work with GPT-4o-Mini-2024-07-18. These modifications ensure that the comparisons are fair by using the same LLM for all methods.

**HLR-SQL.** Since HLR-SQL can autonomously continue querying indefinitely, we impose an upper limit of 25 iterative steps. At the 25th step, HLR-SQL is required to make its final guess, and no further steps are allowed. Additionally, we limit the size of sub-results from sub-queries to 10 rows to not exceed the context limit.

## 4.2 Exp. 1: Varying #Joins

In this experiment, we analyze the accuracies of all baselines and HLR-SQL on the *Spider-HJ* dataset. We break down the results by the number of joins in the SQL query ranging from 2-20 joins. The results of the baselines and HLR-SQL are shown in Figure 2 (a). On the x-axis, we display the number of joins, increasing from left to right. On the y-axis, we display the different execution accuracies.

As shown in Figure 2, the execution accuracy of all baselines decreases significantly with increasing joins counts. This shows the difficulty of state-of-the-art approaches to adapt to the more complex schema. Additionally, we can see that HLR-SQL (green) continuously outperforms all baselines. However, it also shows the same behavior overall, indicating that high join counts are more challenging in general.

## 4.3 Exp. 2: Conversation Length

Since humans take more iterative steps to come to a final SQL for more complex queries, we expect HLR-SQL to do the same. To understand the actual behavior of HLR-SQL, we analyze the conversation length (number of iterative prompts) for each join count, as shown in Figure 2 (b). As can be seen, the average conversation length generally increases with the number of joins. Furthermore, the unexpected performance increase from 7 to 8 joins is also represented in the behavior of HLR-SQL, as we can see a decrease in the average conversation length. This leads us to believe that there are indeed additional characteristics that make these questions easier, even though their number of join operations is higher. The same can be seen for the performance increase from 10 to 16 joins. From this, we conclude that the LLM correctly uses its given capabilities more for harder questions, which aligns with human behavior. However, the overall performance shows that for highly complex queries, further improvements are necessary to fully imitate human behavior and human performance.

## 5 The Road Ahead

In this paper, we examined the impact of complex SQL queries on the Text-to-SQL task. Our experiments have shown that current state-of-the-art approaches have focused too much on queries with low join counts (i.e., less than 4 joins), which is far from realistic scenarios. We argue that in order to solve these queries, a new human-like approach is necessary. Our initial prototype HLR-SQL shows substantial improvements over existing approaches for these queries by mimicking human reasoning and iteratively generating and testing more and more complex queries. However, many open questions remain.

**Real-world data and other dimensions of complexity.** While a high number of joins is certainly a major driver of complexity in real-world SQL queries, they can also be complex due to a variety of other reasons. For instance, queries can be nested or contain complex selection, filter, or group-by expressions. Moreover, real-world data can also be complex, for example by containing ambiguous or cryptic column names and table values. In the future, we thus intend to evaluate our approach on newer datasets such as BEAVER [1] and Spider 2.0 [8] that incorporate these complexities.

**Text-to-SQL as a reasoning task.** Despite the superior performance of HLR-SQL on complex queries of *Spider-HJ*, there are still many queries where it fails, especially with high join counts. Thus, an in-depth error analysis is necessary to understand the failure points of HLR-SQL and when they diverge from human behavior. Moreover, with the announcement of OpenAI o1 and Deepseek R1 [2], reinforcement learning trained reasoning models have shown promising results for reasoning tasks such as mathematical problem solving [2]. Following our argument in this paper, we think that these can be a very good fit for Text-to-SQL as well. However, currently, these models cannot interact with the database during reasoning like HLR-SQL can do, which is crucial to mimic human behavior and generate complex SQL statements, as we have shown in this paper.

**A database interface designed for LLMs.** So far, the interface between HLR-SQL and the database is that HLR-SQL generates SQL sub-queries that are executed on the database. However, for many human-like sub-tasks, SQL on its own is too limited. For example, assuming we want to select all courses in fall, we need to know how *fall* is represented in the database, which could be *fall*, *autumn*, or even a numerical value. This requires a method to look for semantically similar values in the database. Recent approaches solve these issues before SQL generation during schema linking, where they find potentially relevant values from the database and include them in the Text-to-SQL prompt. However, this again results in a static Text-to-SQL pipeline that starts with schema linking and ends with SQL generation. In contrast, we think that schema linking should be part of the reasoning process, and finding values in the database should be realized by extending the database interface for the LLM with a keyword search to search for values semantically similar to "fall." Similarly, we plan to add additional interfaces that the LLM can use during its reasoning process. For example, an interface might be used to automatically detect and return all possible join paths between two tables, such that the LLM can use this join condition.

## Acknowledgments

This work has been supported by the BMBF and the state of Hesse as part of the NHR Program and the HMWK cluster project 3AI. It was also partially funded by the LOEWE Spitzenprofessur of the state of Hesse. We also thank DFKI Darmstadt and hessian.AI for their support.

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

Received 02 April 2025; accepted 29 April 2025