# OpenReview forum: "HLR-SQL: Human-Like Reasoning for Text-to-SQL"
_ACM.org/SIGMOD/2025/Workshop/NOVAS — NOVAS 2025_

### Official Review · Reviewer_nj6v · 2025-04-18

**Confidence:** 4

**Improvement Opportunities:**

- Would be great if we could see how different LLMs stack against each other, specially open source ones
- The main evaluation metric, execution accuracy, warrants elaboration. A small description or pseudo-code of how it is implemented is important to see
- It would be nice to have a couple examples of the dataset to get a better idea of its "shape"
- Many of the shortcomings of this work are already acknowledged, which is great, of course, we would like to see them addressed in future work. The main one is an error analysis with some examples of systematic mistakes which could help point at future directions.
- While the size of the dataset is sufficient, it is still automatically generated and this has an inherent "disconnect" with real data schemas. I understand that there is a tension between scaling a dataset and obtaining high quality, hand-curated, gold data. I think that this work would benefit a lot from an additional benchmark that uses a real data schema and has a few (~100) carefully designed question/query pairs to use as an extrinsic evaluation

**Minor Comments:**

In page 3, second paragraph:
> If the agent select capability (1), (3)

Did you mean _(2), (3)_ instead?

**Short Summary:**

In this paper the authors propose a method for Text-to-SQL called HLR-SQL. The core design choice of the method is its inspiration _human-like reasoning_ analogous to the observation of people following an incremental approach at building complex SQL queries incrementally, growing in complexity at each step.

In order to test the method, they also introduce a dataset named _Spider HJ_ derived procedurally from an existing text-to-sql dataset (Spider). The key difference is that the new dataset requires a significantly larger number of "joins" in the resulting SQL statement in order to successfully answer the input question. The reason for doing this adaptation comes from the observation that real-life scenarios have far more complex data schemas, which translate in a larger number of tables and more complex SQL statements, than those in existing datasets used to benchmark similar approaches.

The experimental results of testing HLR-SQL on Spider HJ show that it is more robust than other commonly used methods across the board regardless of the number of joins necessary for the queries.

Still, the authors acknowledge that there is significant work to address existing challenges on text-to-sql methods using LLMs

**Strong Points:**

- The motivation behind the method, and its explanation is very well introduced and explained, which makes it very easy to appreciate
- The construction of the dataset is documented and relies on an existing established dataset, which is good for replicability. Also, increasing the complexity of the data schema makes the evaluation more similar to real-life scenarios
- The agent architecture is well defined with a fixed "state machine", which helps with the interpretability of the agent's decisions. This. can be appreciated on Figure 2(b).
- The method is benchmarked against baseline methods on the same space, and insight is drawn from the performance of the various methods as the complexity of the query increases

---

### Official Review · Reviewer_RaKh · 2025-04-19

**Confidence:** 4

**Improvement Opportunities:**

W1. Many existing approaches already include schema/data information in context when performing SQL generation. Although this is perhaps not iteratively done nor in a systematic way, the paper should comment  on the benefit of their approach over a one-off use of data/schema characteristics in context. What new information can the LLM uncover by an iterative process?

W2. Would be good to compare/discuss Spider 2.0 https://arxiv.org/abs/2411.07763

W3. Although I like the benchmark, it is not necessarily true that the benchmark represents a real-world workload because the schemas may be broken up in an unrealistic way. It may be harder for an LLM to join back what the broken up schema in the paper. It would be good to have some supporting evidence that this is not the case. For example, showing that the models obtain similar accuracy as spider or bird given the same number of joins

**Minor Comments:**

NA

**Short Summary:**

The paper discusses the problem of how to generate SQL queries from natural language. It proposes an approach for text2sql that attempts to generate the query step by step, and by executing intermediate queries on the database to feeding it back to the llm. The paper also discusses a new benchmark, created based on spider but to include more joins. The paper shows as more joins are considered the accuracy of most approaches drop.

**Strong Points:**

S1. The idea of iteratively constructing the SQL query using input from the database is interesting
S2. The new proposed benchmark is useful for evaluating LLM abilities
S3. The experiments show the benchmark can help provide better understanding of LLM's nl2sql ability

---

### Official Review · Reviewer_ybf6 · 2025-04-21

**Confidence:** 4

**Improvement Opportunities:**

I1: Please provide more details on the approach in Section 2.2. How does the system choose among the four sub-tasks? Is the selection random or learned or llm-based? How is memory used across iterations? Do you include all sub-query results in context? Is there a retrieval or RAG-style mechanism in place?

I2: Include results on standard benchmarks such as Spider 1.0, which has fewer joins. This will provide better context for evaluating the system. Results on BEAVER and Spider 2.0 would significantly improve the paper.

I3: Is the spider-hj benchmark publicly available? If not, would it be released?

I4: Including a discussion section on how this compares to LLMs used for code generation could be useful.

I5: A deeper analysis of the experiments to understand complexity beyond just the number of joins could be useful for both system design and benchmark creation.

**Minor Comments:**

bevior -> behavior in Sec 4.3 Para 1

**Short Summary:**

The paper introduces HLR-SQL a novel method for text-to-sql using llms. They discuss gaps between current benchmarks and real-world sql workloads - they have fewer joins. To address this, the authors propose a new, more complex benchmark (Spider-HJ). They also propose a new iterative autonomous approach to generate sql queries from nl descriptions. Experimental results on the proposed benchmark demonstrate promising improvements over state-of-the-art methods. Overall, the paper addresses a relevant problem, proposes an interesting and well-motivated solution, outlines promising directions for future work, and is well-written. I recommend Accept

**Strong Points:**

S1: The paper studies a relevant problem. It identifies gaps in existing solutions and suggests improvements.

S2: The paper discusses shortcomings of current benchmarks. It introduces a new benchmark with more complex schemas to adress these. The paper also describes a methodology to create such complex schema variants.

S2: The proposed solution is well-motivated and intuitive. Preliminary results show promising improvements.

S3: The future work section outlines interesting and worthwhile directions for further research.